# The 21-Gene Recurrence Score Assay and Prediction of Chemotherapy Benefit: A Propensity Score-Matched Analysis of the SEER Database

**DOI:** 10.3390/cancers12071829

**Published:** 2020-07-08

**Authors:** In Sil Choi, Jiwoong Jung, Byoung Hyuck Kim, Sohee Oh, Jongjin Kim, Jin Hyun Park, Jeong Hwan Park, Ki-Tae Hwang

**Affiliations:** 1Department of Internal Medicine, Seoul Metropolitan Government Seoul National University Boramae Medical Center, Seoul 07061, Korea; hmoischoi@hanmail.net (I.S.C.); jinhyunpak@gmail.com (J.H.P.); 2Department of Surgery, Seoul Medical Center, Seoul 02053, Korea; ciba183@naver.com; 3Department of Radiation Oncology, Seoul Metropolitan Government Seoul National University Boramae Medical Center, Seoul 07061, Korea; karlly71@hanmail.net; 4Medical Research Collaborating Center, Seoul Metropolitan Government Seoul National University Boramae Medical Center, Seoul 07061, Korea; oh.sohee@gmail.com; 5Department of Surgery, Seoul Metropolitan Government Seoul National University Boramae Medical Center, Seoul 07061, Korea; michael5@hanmail.net; 6Department of Pathology, Seoul Metropolitan Government Seoul National University Boramae Medical Center, Seoul 07061, Korea; hopemd@hanmail.net

**Keywords:** breast cancer, chemotherapy, genomic assay, mortality, observational study

## Abstract

Background: To evaluate the performance of the 21-gene recurrence score (RS) assay in predicting chemotherapy benefit in the Surveillance, Epidemiology, and End Results population, we aimed to assess breast cancer-specific mortality (BCSM) by chemotherapy use within each of the RS categories. Methods: Data on breast cancer (BC) cases diagnosed between 2004 and 2015 with available RS results were released. Our analysis included patients with hormone receptor-positive, node-negative early-stage BC (*n* = 89,402), and three RS groups were defined; RS < 11, low; RS 11–25, intermediate; RS > 25, high. A propensity score matched-analysis was performed to assess and compare BCSM. Results: Chemotherapy was significantly associated with a reduced risk of BC death among patients in the high RS group (hazard ratio = 0.782; 95% CI, 0.618–0.990; *p* = 0.041). However, in the low and intermediate RS groups, there were no significant differences in BCSM between patients who received chemotherapy and those who did not. Among those with RS 11–25, chemotherapy benefit varied with tumor size (*p* = 0.001). Conclusions: Our findings provide real-world evidence that the 21-gene RS assay is predictive of chemotherapy benefit among patients in clinical practice. More refined risk estimates would be needed for patients with an intermediate RS.

## 1. Introduction

The 21-gene recurrence score (RS) assay (Oncotype DX, Genomic Health, Redwood City, CA, USA) is one of several commercially-available gene expression assays used to guide treatment decisions in patients with hormone receptor-positive early-stage breast cancer (BC) [1]. The 21-gene RS assay provides prognostic information for patients with early-stage, hormone receptor-positive, node-negative, or node-positive BC who were treated with endocrine therapy alone. Results from multiple validation studies revealed that a high RS (defined as 31 or higher or 26 or higher on a scale of 0 to 100) was associated with a higher rate of distant recurrence and a worse prognosis [2,3,4,5]. Likewise, prospective clinical outcomes studies demonstrated low recurrence rates among patients with a low RS who were treated with endocrine therapy alone [6,7]. Additionally, in several large population-based clinical outcome studies that represented real-world clinical practice, the 21-gene RS was predictive of distant recurrence and breast cancer-specific mortality (BCSM) [8,9,10,11]. The RS based on the 21-gene assay also predicted benefit from adjuvant chemotherapy; patients with a high RS benefited substantially from chemotherapy, whereas chemotherapy had little to no impact on outcomes for patients with a low RS [3,12,13]. Given these findings, the 21-gene RS assay has already had a significant impact on treatment decisions [14,15,16], overall use of chemotherapy [16,17,18], and survival of patients [19]. For patients with an intermediate RS, results from the recent prospective Trial Assigning Individualized Options for Treatment (TAILORx) trial [20], which was designed to test whether chemotherapy is beneficial for patients with hormone receptor-positive, HER2 (human epidermal growth factor receptor 2)-negative, node-negative BC who had a mid-range 21-gene RS (11–25, new intermediate score), revealed that adjuvant endocrine therapy alone was non-inferior to chemo-endocrine therapy in the analysis of invasive disease-free survival (IDFS). Results from a subgroup analysis suggested that chemotherapy may be of some benefit for patients 50 years of age or younger with an RS of 16–25. However, there remains uncertainty regarding the benefit of chemotherapy for most patients who have a mid-range RS.

In this study, to evaluate the performance of the 21-gene RS assay in predicting chemotherapy benefit in the Surveillance, Epidemiology, and End Results (SEER) population, we aimed to assess breast cancer-specific mortality (BCSM) by chemotherapy use within each of the RS categories. We were particularly interested in determining whether chemotherapy benefits patients with an intermediate RS.

## 2. Results

### 2.1. Characteristics of the Study Population

The original primary cohort included a total of 89,402 patients with hormone receptor-positive, node-negative with T1 or T2 BC that had results from the 21-gene RS assay (Appendix A). We noted an overall increase in the use of this assay over time (2004 through 2015; Appendix A). Of these patients, 18,736 (21.0%) had RS < 11, 57,388 (64.2%) had RS 11–25, and 13,278 (14.9%) had RS > 25. The median age was 59 years (range: 18–94 years); 25.9% were ≤50 years old. Most of the patients were white (83.0%). Approximately half (53.1%) had grade 2 tumors, 77.9% had tumors that were ≤2 cm in size, and 73.4% were diagnosed with invasive ductal carcinoma. There were more grade 3 tumors and fewer grade 1 tumors in the high RS group; this group also had fewer ER+PR+ tumors and more ER+PR− tumors than those were found among the lower RS groups. Chemotherapy use was reported as “yes” or “no/unknown”; chemotherapy use (“yes”) increased in proportion to 21-gene RS results with 2.4% (453/18,736) of those with RS < 11, 14.8% (8484/57,388) of those with RS 11–25, and 63.4% (8423/13,278) of those with RS >25 receiving adjuvant chemotherapy. Clinical variables associated with a higher likelihood of receiving adjuvant chemotherapy were younger age at diagnosis (≤50 years), tumor size >2 cm, and high tumor grade (Appendix A).

### 2.2. Propensity Score Matching

Within each of the RS groups, patients were matched by propensity score; there were 444, 8388, and 4495 chemotherapy-treated patients who were matched with the same number of chemotherapy-untreated patients in the low, intermediate, and high RS groups, respectively. All covariates were adequately balanced in the matched cohort, as shown in Table 1. After propensity score matching, our patient cohort included 34.4% who were ≤50 years, 53.6% with grade 2 tumors, and 72.1% with tumors ≤2 cm in size. Median follow-up for all patients was 53 months; 11,449 patients (43.0%) had >5 years of follow-up (RS < 11, 459 (51.7%); RS 11–25, 7669 (45.7%); RS > 25, 3321 (36.9%)).

### 2.3. Chemotherapy Benefit with Respect to RS Group

A total of 465 breast cancer deaths were reported in the matched cohort; 6/888, 178/16,776, and 281/8990 in the groups, including low (RS < 11), intermediate (RS 11–25), and high (RS > 25), respectively (Table 1). KM estimates for the risk of BC death by chemotherapy use for each of the RS groups are shown in Figure 1. A statistically significant benefit from chemotherapy was identified among those in the high RS (>25) group (HR = 0.782; 95% CI, 0.618–0.990; *p* = 0.041; Figure 1c); 5-year and 9-year BCSMs were 3.3% (95% CI, 2.5–4.1%) and 7.3% (95% CI, 5.5–9.1%) for patients who were treated with chemotherapy compared to 4.6% (95% CI, 3.8–5.4%) and 8.4% (95% CI, 6.8–10.0%) for those who were not, respectively. By contrast, there were no statistically significant differences in BCSM between patients who received chemotherapy and those who did not in the low RS (<11) and intermediate RS (11–25) groups (Figure 1a,b). Both patients in the RS < 11 and RS 11–25 groups had lower hazards for BCSM. In the primary unmatched cohort, there were 18,283 patients with RS < 11 who did not receive chemotherapy; for these patients, 5-year and 9-year BCSMs were 0.5% (95% CI, 0.3–0.7%) and 1.2% (95% CI, 0.8–1.6%). Also, for 48,904 patients with RS 11–25 who did not receive chemotherapy, 5-year and 9-year BCSMs were 0.7% (95% CI, 0.5–0.9%) and 2.4% (95% CI, 2.0–2.8%) (Appendix A). Additionally, when we assessed KM estimates for the risk of BC death by chemotherapy use for patients with RS 26–30 (old intermediate-risk and new high-risk category), we identified no significant difference in BCSM between patients who received chemotherapy and those who did not (HR = 0.764; 95% CI, 0.538–1.084; *p* = 0.130, Figure 2); 5-year and 9-year BCSMs were 2.3% (95% CI, 1.5–3.1%) and 5.5% (95% CI, 3.7–7.3%) for patients who received chemotherapy compared to 3.4% (95% CI, 2.4–4.4%) and 7.6% (95% CI, 5.2–10.0%) for those who did not, respectively.

### 2.4. Treatment Interactions Determined by Subgroup Analysis among Patients with RS 11–25

As noted earlier, our findings revealed no apparent benefit from chemotherapy in our evaluation of BCSM among 16,776 patients with RS 11–25. However, we performed a subgroup analysis in order to identify any subgroups that might benefit from chemotherapy. In this cohort, 42.9% were ≤50 years old, 77.0% had grade 1 to 2 tumors, and 72.2% had tumors ≤2 cm in size. A forest plot demonstrating a comparison of BCSM by treatment arm (chemotherapy vs. no chemotherapy) for various covariates is shown in Figure 3. There were no significant interactions between chemotherapy treatment and most of the covariates, including RS subcategory (RS 11–15 vs. RS 16–20 vs. RS 21–25), age (≤50 vs. 50 to 65 vs. >65 years), race (white vs. black vs. others), tumor grade (G1 vs. G2 vs. G3), hormone receptor status (ER+PR− vs. ER+PR+), and adjuvant radiotherapy (yes vs. no). However, we did identify a significant interaction between chemotherapy treatment and tumor size (T1 ≤ 2 cm vs. T2 > 2 cm; *p* = 0.001).

## 3. Discussion

In this study, we investigated 89,402 patients with hormone receptor-positive, node-negative early-stage BC treated based on the 21-gene RS results as applied to a large real-world population-based registry. We found that the distribution of RS ranges using the TAILORx cutoff points included a low of 21.0%, an intermediate of 64.2%, and a high of 14.9%. This RS distribution was quite similar to that observed in the TAILORx trial; in which approximately 70% of the patients had a mid-range RS [6,20,21]. This study aimed to evaluate the benefit of chemotherapy by estimating the risk of BC death by chemotherapy use within the TAILORx-defined RS groups. Among the 8890 patients in a propensity score-matched cohort all with hormone receptor-positive, node-negative BC and a high RS of >25, we identified a significant benefit from chemotherapy (HR = 0.782; 95% CI, 0.618–0.990, *p* = 0.041); this finding reconfirmed the evidence that adjuvant chemotherapy was beneficial in this group of patients [3,12,13]. By contrast, our findings revealed no apparent benefit from chemotherapy among those in the RS < 11 and RS 11–25 groups; in fact, clinical outcomes were overall very good in all patients with an RS < 26. For chemotherapy-untreated patients in the primary unmatched cohorts, KM estimates for the risk of BC death at 5 and 9 years were 0.5% and 1.2% in patients with RS < 11 and 0.7% and 2.4% in patients with RS 11–25, respectively. Our results were similar to those observed in the TAILORx trial and likewise in several population-based studies [6,8,10,20,22]. The TAILORx prospective randomized trial included 6711 patients with a mid-range RS of 11–25, and in this group, adjuvant endocrine therapy alone was non-inferior to chemo-endocrine therapy; this finding provides evidence that adjuvant chemotherapy was not beneficial for these patients [20]. Furthermore, in a population-based study using a large prospectively designed registry, no significant differences in 5-year distant recurrence risk by chemotherapy use among those in the intermediate risk RS groups (RS 18–25 and RS 26–30) [10]. Recently, a SEER population-based study revealed that 9-year BCSM was <4% among patients with RS 0–25 and node-negative disease who did not receive chemotherapy, and for RS 26–100, 9-year BCSM was lower among those who received chemotherapy than those who did not [22].

Of note, in our analysis, in the RS 26–30 group, there was no statistically significant difference in BCSM between patients who were treated with chemotherapy and those who were not (Figure 2). Patients categorized by the old intermediate score of RS 26–30 had not been included in the TAILORx prospective randomized trial for a mid-range RS of 11–25 [20]; however, in a recent secondary analysis of the TAILORx trial that included 1389 patients with RS 26–100 assigned to adjuvant chemo-endocrine therapy [21], 546 patients (42%) with RS 26–30 had better outcomes than did the remaining patients with RS 31–100. Currently, the benefit from chemotherapy for those with RS 26–30 remains uncertain, as presented in the National Comprehensive Cancer Network [23] and American Society of Clinical Oncology [24] clinical practice guidelines. Future studies will be needed to provide coherent guidelines for this group of patients.

As shown in the TAILORx trial and also here in our study, most BC patients evaluated fell within the mid-range RS of 11–25, and chemotherapy did not provide a meaningful improvement in outcomes for these patients overall. However, reports from several studies suggested that clinical parameters, when added to the 21-gene RS, might provide improved prognostic value [20,25,26]. The RS-pathology-clinical (RSPC) risk assessment, which integrated RS with clinicopathological measures (age, tumor size, grade, type of endocrine therapy) added significant prognostic information to the 21-gene RS, and could aid physicians in making treatment decisions, especially in patients with an intermediate RS; interestingly, many (77.9%) of the 272 patients classified as intermediate risk by RS were reassigned to high risk (16.9%) or to low risk (55.1%) by the RSPC classifier [25]. In the TAILORx subgroup analysis, some benefit of chemotherapy for IDFS was identified in younger patients (≤50 years) with a RS of 16–25 [20]; in addition, clinical-risk stratification based on the tumor size and histologic grade added prognostic information to the 21-gene RS among women of ≤50 years who had received endocrine therapy alone [26]. However, the subgroup analysis in TAILORx was not preplanned and retrospective, and there was low distant recurrence event rate for the study overall. Thus, the findings from the subgroup analysis require further validation. As shown in major international guidelines incorporating results from TAILORx, age can be taken into account when making decisions about chemotherapy, but this variable should not be a major determinant in decision-making [23,24]. In our subgroup analysis, findings diverged somewhat from those of the TAILORx trial. For example, we did not find a significant interaction between chemotherapy treatment and age when evaluating BCSM. However, we did identify a significant interaction between chemotherapy treatment and tumor size (*p* = 0.001), which suggested some survival benefit from chemotherapy in patients who had tumors >2 cm in size. Considering that 73.9% had clinical low risk tumors in the TAILORx trial [20] (risk was defined as in the Microarray in Node Negative Disease May Avoid Chemotherapy (MINDACT) trial [27]), the benefit from chemotherapy for clinical high risk patients with a mid-range RS needs to be evaluated further. In fact, in other genomic assays, tumor size is integrated with molecular score to estimate recurrence risk [28,29]. More refined risk estimates to identify patients who could benefit from more effective therapy would be needed, specifically in tumors with a mid-range RS.

This analysis of data from real-world population-based large registries provides valuable information on clinical outcomes of patients that is relevant in actual clinical practice. Of note, this is the largest-to-date study of the performance of the 21-gene assay in predicting chemotherapy benefit. Nonetheless, this study has several limitations. First, the SEER program is a population-based registry, not clinical trial, and thus, the patients were not randomized to treatment. As such, to reduce the effect of a selection bias and confounding factors, we performed a propensity score matched-analysis to evaluate the benefit of chemotherapy. Second, the SEER program collects no information on breast cancer recurrence and progression, and as such, these outcomes could not be evaluated. Third, we reported BCSM and its association with adjuvant chemotherapy in our study; unfortunately, chemotherapy use is known to be under-reported in SEER [30], and thus, the results need to be interpreted with caution. Lastly, at the time that this analysis was performed, follow-up time for this patient cohort was relatively short, and longitudinal follow up is warranted for more reliable information on long-term survival outcomes.

## 4. Materials and Methods

### 4.1. Data Source

The SEER program of the National Cancer Institute is a population-based cancer registry program that was initiated in 1973. The SEER program has collected 21-gene RS assay results for all breast cancer cases diagnosed since 2004. Data on breast cancer cases diagnosed between 2004 and through 2015 that were linked to the 21-gene assay RS results were released for this study following SEER approval. All data and parameters were utilized from SEER Program Research Data (1973–2015) and specialized data of Oncotype DX released in April 2018, based on the November 2017 submission.

### 4.2. Patient Selection

We identified a total of 710,217 cases that were newly-diagnosed with adult female breast cancer between the years 2004 and 2015 from the SEER registries (Appendix A). Of these, we identified 110,909 cases associated with results of the 21-gene assay; among these cases, 107,930 underwent primary surgery for invasive non-metastatic disease; of these, we selected 89,402 patients with hormone receptor-positive (estrogen receptor [ER] and/or progesterone receptor [PR]-positive), node-negative patients with T1 or T2 who were followed for a period of ≥1 month. We abstracted data on demographics, tumor characteristics, treatment modalities, and survival outcomes. We used RS cutoff points employed in the TAILORx trial, including low (RS < 11), intermediate (11–25), and high (>25).

### 4.3. Propensity Score Matching

As there is likely to be a selection bias inherent in use of this assay and also a bias for choice of therapy, propensity score adjustment was utilized within each of the RS groups. We estimated the propensity score or the probability of receiving adjuvant chemotherapy using a multivariate logistic regression model. Covariates included year of diagnosis, age at diagnosis, race, T category, tumor grade, hormone receptor status, and use of radiotherapy. A 1:1 matching was then performed by using the nearest neighbor method with a caliper width less than 0.25 standard deviations using the MatchIt’ package in R (version 3.5.2; http://www.r-project.org). We examined the balance in the baseline covariates in the matched data by using standardized mean differences.

### 4.4. Statistical Analysis

We compared the baseline categorical variables according to chemotherapy status within each RS risk group using the chi-square test in both the matched and unmatched cohort. We estimated the risk of BC death in the matched and unmatched cohort using Kaplan–Meier (KM) curves. In the matched cohort, we compared BCSM among patients who received chemotherapy compared to those who did not by the log-rank test, and hazard ratios (HRs) were calculated using Cox proportional hazard regression model. All tests were two-sided, and *p*-values less than 0.05 were considered significant. Statistical analysis was performed using IBM SPSS software (version 20.0; IBM Corp., Armonk, NY, USA) and SAS software (version 9.3; SAS Institute, Cary, NC, USA).

## 5. Conclusions

Our SEER study results provide real-world evidence for the prognostic and predictive value of the 21-gene RS assay, specifically with respect to predicting the chemotherapy benefit among patients with hormone receptor-positive, node-negative BC in clinical practice. As noted, for patients with a mid-range RS, clinical factors could add prognostic information to the 21-gene RS and provide more information toward improving treatment-related decisions.

## Figures and Tables

**Figure 1 cancers-12-01829-f001:**
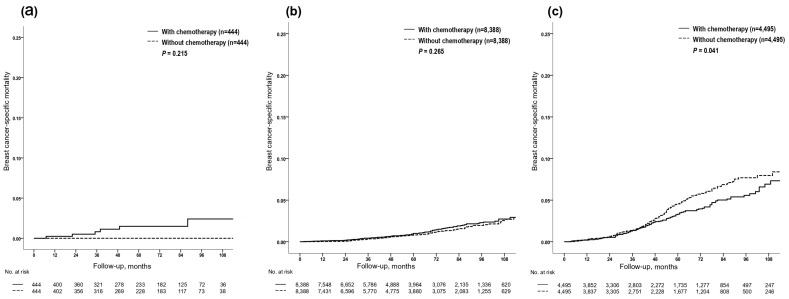
The Kaplan–Meier estimates of the risk of BC death according to chemotherapy by RS risk groups (<11, 11–25, >25) in the matched cohort. (**a**) low-risk, RS < 11; (**b**) intermediate-risk, RS 11–25; (**c**) high risk, RS > 25. Abbreviations: BC, breast cancer; RS, recurrence score.

**Figure 2 cancers-12-01829-f002:**
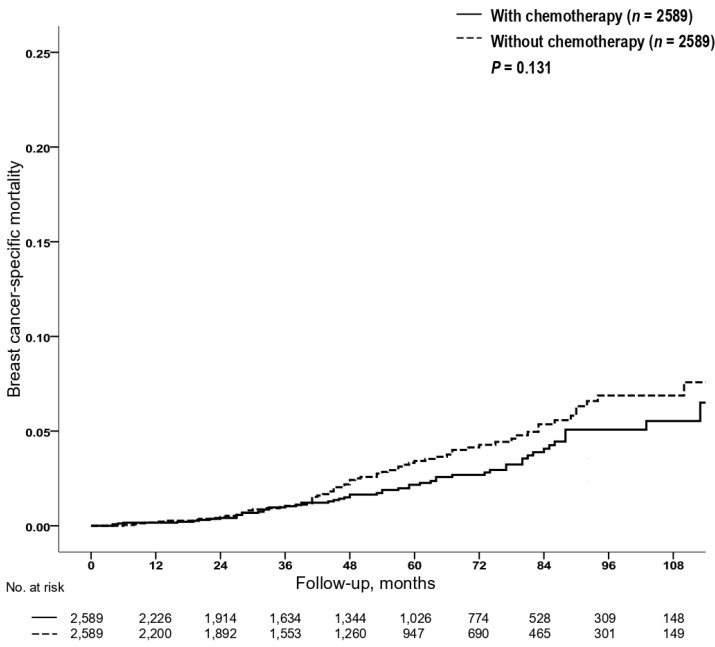
The Kaplan–Meier estimates of the risk of BC death according to chemotherapy in the RS 26–30 group. Abbreviations: BC, breast cancer; RS, recurrence score.

**Figure 3 cancers-12-01829-f003:**
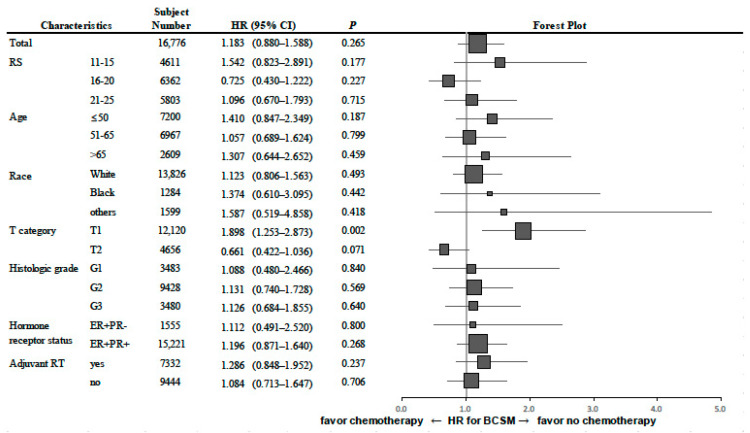
Forest plot demonstrating a comparison of BCSM by treatment arm for chemotherapy vs. no chemotherapy for different subgroups among patients with RS 11–25. Abbreviations: BCSM, breast cancer specific mortality; RS, recurrence score.

**Table 1 cancers-12-01829-t001:** Baseline characteristics according to chemotherapy in the matched cohort.

Characteristic	RS 0–10		*p*	RS 11–25		*p*	RS 26–100		*p*
	No Chemotherapy	Chemotherapy		No Chemotherapy	Chemotherapy		No Chemotherapy	Chemotherapy	
	(*n* = 444)	(*n* = 444)		(*n* = 8388)	(*n* = 8388)		(*n* = 4495)	(*n* = 4495)	
**Year**			0.924			0.969			1
2004–2006	42 (9.5%)	38 (8.6%)		738 (8.8%)	720 (8.6%)		304 (6.8%)	304 (6.8%)	
2007–2009	161 (36.3%)	164 (36.9%)		2638 (31.4%)	2639 (31.5%)		1130 (25.1%)	1129 (25.1%)	
2010–2012	129 (29.1%)	135 (30.4%)		2654 (31.6%)	2661 (31.7%)		1508 (33.5%)	1508 (33.5%)	
2013–2015	112 (25.2%)	107 (24.1%)		2358 (28.1%)	2368 (28.2%)		1553 (34.5%)	1554 (34.6%)	
Patient age, years (range)			0.946			0.988			1
≤50	189 (42.6%)	187 (42.1%)		3599 (42.9%)	3601 (42.9%)		801 (17.8%)	802 (17.8%)	
>50	255 (57.4%)	257 (57.9%)		4789 (57.1%)	4787 (57.1%)		3694 (82.2%)	3693 (82.2%)	
Race			0.888			0.87			1
White	362 (81.5%)	356 (80.2%)		6919 (82.5%)	6907 (82.3%)		3759 (83.6%)	3759 (83.6%)	
Black	34 (7.7%)	39 (8.8%)		633 (7.5%)	651 (7.8%)		391 (8.7%)	391 (8.7%)	
Others	45 (10.1%)	47 (10.6%)		805 (9.6%)	794 (9.5%)		339 (7.5%)	339 (7.5%)	
Unknown	3 (0.7%)	2 (0.5%)		31 (0.4%)	36 (0.4%)		6 (0.1%)	6 (0.1%)	
T category			0.944			0.85			1
T1	283 (63.7%)	281 (63.3%)		6066 (72.3%)	6054 (72.2%)		3262 (72.6%)	3262 (72.6%)	
T2	161 (36.3%)	163 (36.7%)		2322 (27.7%)	2334 (27.8%)		1233 (27.4%)	1233 (27.4%)	
Histologic type			0.184			0.168			0.49
IDC	301 (67.8%)	325 (73.2%)		6030 (71.9%)	6149 (73.3%)		3668 (81.6%)	3672 (81.7%)	
IDC + ILC	30 (6.8%)	32 (7.2%)		695 (8.3%)	687 (8.2%)		275 (6.1%)	260 (5.8%)	
ILC	49 (11.0%)	42 (9.5%)		1040 (12.4%)	968 (11.5%)		307 (6.8%)	290 (6.5%)	
Others	64 (14.4%)	45 (10.1%)		623 (7.4%)	584 (7.0%)		245 (5.5%)	273 (6.1%)	
Histologic grade			0.921			0.989			1
1	132 (29.7%)	129 (29.1%)		1747 (20.8%)	1736 (20.7%)		426 (9.5%)	426 (9.5%)	
2	240 (54.1%)	242 (54.5%)		4707 (56.1%)	4721 (56.3%)		2188 (48.7%)	2188 (48.7%)	
3	56 (12.6%)	60 (13.5%)		1739 (20.7%)	1741 (20.8%)		1824 (40.6%)	1825 (40.6%)	
Unknown	16 (3.6%)	13 (2.9%)		195 (2.3%)	190 (2.3%)		57 (1.3%)	56 (1.2%)	
HR status			1			0.831			1
ER+PR−	3 (0.7%)	4 (0.9%)		773 (9.2%)	782 (9.3%)		1193 (26.5%)	1194 (26.6%)	
ER+PR+	441 (99.3%)	440 (99.1%)		7615 (90.8%)	7606 (90.7%)		3302 (73.5%)	3301 (73.4%)	
Radiation therapy			1			1			1
No	195 (43.9%)	194 (43.7%)		3666 (43.7%)	3666 (43.7%)		2405 (53.5%)	2406 (53.5%)	
Yes	249 (56.1%)	250 (56.3%)		4722 (56.3%)	4722 (56.3%)		2090 (46.5%)	2089 (46.5%)	
Median follow-up, months (range)	62 (30–85)	62 (33–86)		55 (28–83)	57 (29–84)		47 (22–74)	48 (22–76)	
Deaths									
breast cancer	0	6 (1.4%)		81 (1.0%)	97 (1.2%)		156 (3.5%)	125 (2.8%)	
other cause	14 (3.2%)	10 (2.3%)		198 (2.4%)	134 (1.6%)		185 (4.1%)	105 (2.3%)	

Abbreviations: ER, estrogen receptor; HR, hormone receptor; IDC, invasive ductal carcinoma; ILC, invasive lobular carcinoma; PR, progesterone receptor; RS, recurrence score.

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
