# Peer review of "The 21-Gene Recurrence Score Assay and Prediction of Chemotherapy Benefit: A Propensity Score-Matched Analysis of the SEER Database"

_cancers, 2020, doi:10.3390/cancers12071829_

Round 1

Reviewer 1 Report

Very good study that provides " real world" validation of a widely used genomic assay for decision making in treatment of early stage ER (+) breast cancer. Large patient population, valid outcomes and good statistical analysis. Discussion does highlight potential drawbacks to study design.

This study addresses the “real world”  use of a 21 gene recurrence score assay to help determine chemotherapy benefit in early stage ER(+) breast cancer. It is a retrospective analysis of the SEER population, looking at almost 90,000 patients treated between 2004-2015.

This genomic assay has been studied in numerous other retrospective and prospective analyses. However the current paper is original in the very large number of patients studied over a long follow-up period in the general community.

This study adds important confirmatory information from a very large cohort in the  “real-world” setting to prior data.  I feel that this is very helpful  to physicians in community practice globally and reiterates the usefulness of the 21 gene assay in treatment decision making.

The text is clear and easy to read. The evidence and arguments are consistent with the data presented. The discussion addresses the main question posed about the predictive ability of the 21 gene assay in determining which early breast cancer cases should be treated with chemotherapy in addition to hormonal therapy. The study confirms that the genomic assay is a valid tool to predict chemotherapy benefit in the appropriate high risk subset of patients. 

Author Response

Thank you for your valuable comments and positive response to our manuscript.

Reviewer 2 Report

This is a propensity score-matched analysis using SEER breast cancer cohort (2004-2015) to evaluate the predictive value of oncotype score. The primary end point of interest is breast cancer specific mortality (BCSM). In consistence with the TAILORx study and others, the authors observed a significant benefit of chemo-therapy on in the high RS cohort only. This topic has been studied in multiple studies (as cited by the authors), however, it was not clear to me the significance of the current study. It will be informative to have some further discussion in comparing the results of prior studies with the aid of a table. The study design and statistical analysis were properly carried out. On the analysis of subgroup with RS between 26 and 30, the author claimed that there is no significant difference. However, if you take a look at the plot, it is very similar to the high RS group and the HR estimate is very close. It makes one wonder if it is simply due to the number of events. It will be informative to present also a subgroup analysis of RS >30.

Author Response

Thank you for your valuable comments and suggestions.

We answered to your questions and please see the attachment.

Reviewer 3 Report

This is a well-written manuscript and recommend that it be published.

The 21-gene recurrence score assay provides prognostic information for distant recurrence in hormone–receptor-positive, ERBB2-negative early breast cancer that is independent of clinicopathologic features, and is also predictive of adjuvant chemotherapy benefit when the recurrence score is high.  Assessing the breast-cancer genes related to proliferation and invasion of very high importance.  This describes this manuscript’s focus on determining whether chemotherapy benefits patients that possess an intermediate resistance score.

This is a very large and significant study that includes a sizable population of patients that possess hormone receptor-positive, node-negative with TNM T1 or T2 breast cancer from recurrent score assays.  The “Materials and Methods” section is well written being both brief and to the point.

The discussion section is also well written, encompassing notations of important findings with appropriate interpretations. That the recurrence score group of 26-30 demonstrated no statistically significant difference in breast cancer specific mortality between patients who were treated with chemotherapy with those who were not, is an important finding. That chemotherapy did not provide meaningful improvement in outcomes for patients with recurrence score of 11-25, is also very noteworthy. However, the fact that within a subgroup analysis there was an identification of a significant interaction between chemotherapy treatment and tumor size suggesting some survival benefit from chemotherapy in patients who had tumors greater than 2 cm in size, is an important finding.

All in all, as noted, above, this is an exceptionally well-written paper presenting important findings based upon a significant data pool.

Author Response

(The authors gave the same response as above.)

Reviewer 4 Report

Authors reevaluate breast cancer recurrence score using oncotype DX. This research data is obvious in clinical setting, however authors strength is including huge sample size derived from SEERS and performed statistics using propensity score matching.

This manuscript is well written by proper English, and I do not feel any revision. We may accept or not accept owing to the priority of this manuscript.

Author Response

(The authors gave the same response as above.)
